# Responses of Six Wheat Cultivars (*Triticum aestivum*) to Wheat Aphid (*Sitobion avenae*) Infestation

**DOI:** 10.3390/insects13060508

**Published:** 2022-05-27

**Authors:** Ke-Xin Zhang, Hong-Yan Li, Peter Quandahor, Yu-Ping Gou, Chun-Chun Li, Qiang-Yan Zhang, Inzamam Ul Haq, Yue Ma, Chang-Zhong Liu

**Affiliations:** 1Biocontrol Engineering Laboratory of Crop Diseases and Pests of Gansu Province, College of Plant Protection, Gansu Agricultural University, Lanzhou 730070, China; zhangkx199404@163.com (K.-X.Z.); gouyp@gsau.edu.cn (Y.-P.G.); lichch1992@163.com (C.-C.L.); zhangqiangyan2@163.com (Q.-Y.Z.); inzamam@st.gsau.edu.cn (I.U.H.); my995654123@163.com (Y.M.); 2Wuwei Shiyanghe Forestry General Field, Wuwei 733000, China; lihongyan1224@163.com; 3CSIR-Savanna Agricultural Research Institute, Tamale P.O. Box TL 52, Ghana; quandooh@yahoo.com

**Keywords:** wheat, *Sitobion avenae*, aphid resistance, biological parameters, total phenol, flavonoid

## Abstract

**Simple Summary:**

*Sitobion avenae* Fabricius is an important wheat aphid species in China, causing significant losses to wheat production. Improving host-plant resistance is an effective and environmentally friendly method of aphid control. *Sitobion avenae* resistance and the total phenolic and flavonoid content accumulation of six wheat cultivars to *S. avenae* infestation were investigated to elucidate responses of six wheat varieties against *S. avenae*. Among the six tested wheat cultivars, Yongliang No.15 and Ganchun No.18 demonstrated high resistance to *S. avenae*. The correlation analysis revealed a positive relationship between total phenol and flavonoid content accumulation and developmental duration (DD), and a negative relationship between accumulation and weight gain (WG) and mean relative growth rate (MRGR). The correlation between flavonoid and biological parameters was statistically stronger than total phenol. Our findings could serve as a theoretical basis for further research into the resistance mechanism of wheat varieties to *S. avenae*.

**Abstract:**

Resistant variety screening is widely recommended for the management of *Sitobion avenae*. The purpose of this study was to assess responses of six wheat varieties (lines) to *S. avenae*. The aphid quantity ratio (AQR) was used to assess *S. avenae* resistance. Pearson’s correlation coefficient was used to perform a correlation analysis between AQR, biological parameters, and the accumulation of total phenolic and flavonoid content. When compared to the other cultivars, the results showed that two cultivars, Yongliang No.15 and Ganchun No.18, had high resistance against *S. avenae*. The correlation analysis revealed a positive relationship between total phenol and flavonoid content accumulation and developmental duration (DD), and a negative relationship between accumulation and weight gain (WG) and mean relative growth rate (MRGR). The correlation between flavonoid and biological parameters was statistically stronger than the correlation between total phenol and biological parameters. This research provides critical cues for screening and improving aphid-resistant wheat varieties in the field and will aid in our understanding of the resistance mechanism of wheat varieties against *S. avenae.*

## 1. Introduction

Wheat (*Triticum aestivum* L. (Poaceae)) is the world’s third most important food crop, trailing only rice (*Oryza sativa* L.), and maize (*Zea mays* L.). It is an important food source because its production accounts for 17 percent of the world’s arable land, supplying food for nearly half of the world’s population [1]. Over 700 million tons of wheat are produced annually on approximately 220 million hectares of land worldwide [2,3]. However, coincident with increasing concentration of carbon dioxide (CO_2_) has been the primary limiting factor played by the wheat aphid in wheat yield and quality [4]. Aphids are significant economic pests on cultivated plants in agriculture worldwide [5]. They are piercing–sucking pests of several cereal crops that cause severe damage through sucking nutrients, deposition of honeydew, and transmission of viral disease [6]. Because of the severity of drought stress, the English grain aphid, *Sitobion avenae* Fabricius (Hemiptera: Aphididae), is a cosmopolitan economic pest that threatens wheat production in Northwestern China [7,8,9]. *Sitobion avenae* damage affects approximately 13 million hectares per year and reduces wheat yield by 40% in China [10].

The management of *S. avenae* has recently become a longstanding issue due to the emergence of biotypes that are resistant to a wide range of host-plant resistance traits in common wheat and populations that are resistant to a diverse array of insecticides [11]. The scientific community has agreed that breeding and growing aphid-resistant cultivars through plant resistance is an important strategy and primary task for aphid management [12]. Host-plant resistance not only affects herbivore growth and development, but it also changes the composition and content of chemical substances in herbivores, which affects natural enemies that feed on them [13]. Plants have developed and evolved a considerable variety of sophisticated defensive strategies to thwart and escape various insect attacks for over 350 million years of coevolution between plants and insects [14]. In response to insect feeding, host plants can induce signal transduction, which then activates the corresponding physiological and biochemical reactions. Host-plant resistance is manifest as the chemical and physical mechanisms by which plants defend against pest attack through antibiosis (reduced aphid growth or fecundity), antixenosis (aphid nonpreference), tolerance (plant compensatory growth after aphid feeding), and/or combinations of these categories of resistance [15]. The mechanism can also be categorized into constitutive and induced resistance [16].

Constitutive resistance is demonstrated autonomously of the attack, whereas the induced resistance occurs directly when the plant is under stress as a result of the attack [16,17]. Secondary metabolites in plants are natural defense mechanisms against pathogenic invasion and herbivores and insect attacks [17,18,19]. Terpenes (monoterpenes, diterpenes, and triterpenes), phenol (coumarin, lignin, and flavonoid), and nitrogen-containing secondary metabolites (alkaloids, cyanogenic, glucosides, and nonprotein amino acids) are prominent examples of these [20]. It is now well established from studies that flavonoid and simple phenol are dominant plant secondary metabolites that protect plants from insect pest attack [21,22,23,24]. High phenol and flavonoid content in alfalfa-influenced pea aphid (*Acyrthosiphon pisum*) larval development, reproduction period, and fecundity [25].

Improving natural resistance in plant cultivars and pest management has emerged as a major goal of scientific investigation. The current economic conditions and production systems confronting Chinese wheat farmers make insecticidal control of aphids a less viable option; thus, cultivar resistance is of paramount importance. This has the potential to significantly contribute to food and income security, alleviate poverty, and reduce farmer risk in vulnerable agricultural environments. This will also help to reduce the escalation of problems associated with chemical control strategies, such as insecticide resistance, market demand, and environmental contamination. Much research has been performed on the relationship between secondary substances and aphid resistance [21,22,23,24,25,26,27,28]. To date, research on aphid-resistance resources is extremely scarce. This study was conducted to screen the resistance of six wheat cultivars against *S. avenae* and explore the resistance mechanisms.

## 2. Materials and Methods

### 2.1. Plants and Insects

The experiment was carried out at Gansu Agricultural University’s Insect Ecology Laboratory in Lanzhou, China (36°5′ N, 103°41′ E, 1530 m above sea level). Longzhong No.3, Huining No.21, Ganchun No.29, Longchunza No.2, Yongliang No.15, and Ganchun No.18 were the wheat varieties used in the experiment. Seeds of these six varieties were obtained from Gansu Agricultural University’s Agronomy College in China. Prior to sowing, the seeds were surface disinfected by soaking for 10 min in a 5 percent (*m*/*v*) sodium hypochlorite solution and then thoroughly washing with sterile distilled water for 1 min. The seeds were germinated in Petri dishes lined with moist filter paper in the dark at a temperature of 25 ± 1 °C. The uniformly germinating seeds were transplanted into soil-filled plastic pots (10 cm × 10 cm × 9 cm). The plastic pots were kept in an artificial intelligence illumination incubator (Shanghai Yuejin Medical Instruments Co., Ltd., Shanghai, China) at 25 ± 1 °C, 50% ± 10% relative humidity, and 14 h illumination. The grain aphids, *Sitobion avenae* Fabricius, were collected from a winter wheat field at the experimental field of the Gansu Agricultural University and reared on wheat seedlings (variety: Xinong 3517) in an artificial intelligence illumination incubator, controlled at temperature 25 ± 1 °C and photoperiod 14:10 (L:D), for ten generations.

### 2.2. Aphid Resistance Evaluation in the Laboratory

Plump wheat seeds of various varieties were chosen and planted in plastic pots, two pots for each variety and ten seeds per pot. On the seventh day after planting, 10 seedlings were chosen from each pot. Each seedling was infested with five nymph aphids. The infested seedlings were covered with gauze nets to keep the aphids at bay. Each variety received a total of 20 wheat plants, and each variety was replicated three times. To keep the soil moist, all pots were watered evenly. The number of aphids colonized and the number of aphids were determined 7 and 13 d after infestation, respectively.

The aphid quantity ratio (AQR) method was used to evaluate the resistance of the selected varieties (lines) to *S. avenae*. The aphid quantity ratio (AQR) was calculated as:

AQR = Average number of aphids per plant of a given cultivar (line)/Average number of aphids per plant for all observed wheat cultivars (lines).

The wheat resistance level was identified as described by Painter [29] and previously used in other studies [30,31]. Aphid resistance level is categorized into 7 grades (Table 1), which include: immune (I), highly resistant (HR), moderately resistant (MR), low resistant (LR), low susceptible (LS), moderately susceptible (MS), and highly susceptible (HS).

### 2.3. Aphid Growth and Development

The experiment was carried out in an artificial intelligence illumination incubator with a photoperiod of 16 h: 8 h (L: D), a temperature of (25 ± 1) °C, and a humidity of (50% ± 10%) RH. The single-head aphids and single-dish feeding method were used in this experiment. Each Petri dish (90 mm in diameter and 15 mm in height) contained a moisten filter paper and a fresh wheat leaf. Throughout the experiment, the leaves were replaced on a daily basis. Wheat leaves were collected from wheat plants at the same stage of development. A single nymph was carefully weighed and placed on each Petri dish (within 24 h of reproduction). W_1_ was the initial weight recorded (10 aphids per group were weighed to minimize possible errors). In total, 90 replicates of each treatment were used in the experiment. Molting and aphid survival were monitored and recorded at regular intervals throughout the day until they reached adulthood. Each adult aphid’s weight was reweighed and recorded as W_2_. The following formulas were used to calculate the developmental duration (DD), weight gain (WG), and mean relative growth rate (MRGR) [32]:

Developmental duration (DD) refers to the time from birth to adult emergence
(1)$$\mathrm{Weight}\;\mathrm{gain}\;(\mathrm{WG})=W_{2}-W_{1}$$
(2)$$\mathrm{Mean}\;\mathrm{Relative}\;\mathrm{Growth}\;\mathrm{Rate}\;(\mathrm{MRGR})=(\ln\;W_{2}-\ln\;W_{1})/\mathrm{DD}$$
where W_2_ and W_1_ are the adult weight, and first-instar nymph weight (newly born). An electronic balance (Sartorius MSE3.6P-0CE-DM, Göttingen, Germany) was used to weigh the aphids.

### 2.4. Total Phenol Compounds Content Assay

Preparation of extracts: 0.5 g fresh leaves from each treatment were accurately weighed and a small amount of quartz sand and 2 mL deionized water were added. The leaves were ground in an ice bath before being transferred to a 10 mL test tube; 5 mL of deionized water was added, mixed, and filtered through four layers of gauze. Filtrate was kept at 4 °C in a constant volume of 5 mL.

The total phenol content of wheat cultivar was determined using the colorimetric Folin–Ciocalteu reagent method, as previously described by Chlopicka [33], with appropriate modifications. Briefly, 0.3 mL extracts were diluted with 2.7 mL of deionized water, and oxidized by adding 0.15 mL Folin-Ciocalteu reagent. After 3 min, the solution was neutralized with 0.3 mL of 10% sodium carbonate. After 60 min, the absorbance of the mixture was measured at 700 nm using an ultraviolet spectrophotometer with gallic acid as a standard. The total phenol content was expressed in micromoles of gallic acid equivalent (GAE) per gram of grain. Data were reported as mean ± SD for at least three replications.

### 2.5. Total Flavonoid Content Assay

The total flavonoid content was assayed using the method described by Chlopicka [33], with minor modifications. In brief, appropriate dilutions of sample extract were reacted with 5% sodium nitrite (NaNO_2_); after standing for 5 min, 10% AlCl_3_·6H_2_O was added. After 6 min, 1 mol/L NaOH was added in a fixed volume of distilled water. The solution’s absorbance at 510 nm was immediately measured with an ultraviolet spectrophotometer and compared to known catechin concentration standards. Flavonoid content was expressed as milligrams of catechin equivalents per 100 g dry weight. Data were reported as mean ± SD for at least three replications.

### 2.6. Statistical Analysis

Data were expressed as means and standard deviation (SD) for at least three measurements for each cultivar. All statistical analyses and graphing were performed by OriginPro 2021 software (OriginLab Corp). Normality of data distributions was tested using the Shapiro–Wilk normality test. Equality of variance between groups was tested using the Levene test for homogeneity of variance. The AQR, biological parameters, total phenol, and flavonoid content of each cultivar were tested by one-way ANOVA followed by Tukey’s multiple comparisons test with a 95% confidence interval of the difference. Correlation analysis was performed by Pearson correlation coefficient calculation. Pearson correlation coefficients and *p*-values were used to show correlations and their significance. Values of *p* < 0.05 were considered significant.

## 3. Results

### 3.1. Evaluation of Wheat Resistance to S. avenae

AQR was used to assess the level of resistance to *S. avenae* in six wheat varieties/lines. AQR was defined as the average number of aphids on each cultivar divided by the average number of aphids on all cultivars. Longchunza No.2 differed significantly from the four cultivars Longzhong No.3, Huining No.21, Yongliang No.15, and Ganchun No.18 (F_4,10_ = 6.19, *p* < 0.05). Longchunza No.2 had a significantly higher (*p* < 0.05) AQR than Longzhong No.3, Huining No.21, Yongliang No.15, and Ganchun No.18. There was also a statistically significant difference (F_3,8_ = 12.36, *p* < 0.05) between Ganchun No.29 and Huining No.21, Yongliang No.15, and Ganchun No.18. Ganchun No.29 had a higher AQR than Huining No.21, Yongliang No.15, and Ganchun No.18 cultivars. However, no statistically significant difference was found between Ganchun No.29 and Longchunza No.2 (F_1,4_ = 0.005, *p* > 0.05). Differences between four wheat cultivars, Longzhong No.3, Huining No.21, Yongliang No.15, and Ganchun No.18, were also not statistically significant (F_3,8_ = 0.38, *p* > 0.05). Longchunza No.2 had the highest AQR of the six cultivars (1.281). Yong Liang 15 had the lowest AQR (0.893). The findings revealed that the wheat varieties/lines had a discernible effect on the AQR (Figure 1). The preliminary results also showed that Yongliang No.15 exhibited low resistance (LR) to *S. avenae*; the three cultivars Longzhong No.3, Huining No.21, and Ganchun No.18 were low susceptible (LS), and two cultivars of Ganchun No.29 and Longchunza No.2 were moderately susceptible (MS). Highly susceptible (HS), moderately resistant (MR), and highly resistant (HR) cultivars were not observed among the six cultivars in this study.

### 3.2. Biological Parameters of S. avenae on Different Wheat Varieties

Different wheat varieties had an effect on the biological parameters of *S. avenae*. More specifically, the developmental duration (DD) of *S. avenae* differed between wheat varieties/lines. Ganchun No.18 had the longest DD of *S. avenae* (6.83 d), while Longchunza No.2 had the shortest (6.19 d). There was a significant difference in DD of *S. avenae* between Ganchun No.18 and Longchunza No.2 (F_1,4_ = 7.91, *p* < 0.05). However, no significant difference (F_3,8_ = 0.52, *p* > 0.05) in DD of *S. avenae* occurred between the remaining four varieties: Longzhong No.3, Huining No.21, Ganchun No.29, and Yongliang No.15 (Table 2).

Weight gain (WG) is the weight difference between a nymph and an adult aphid. The WG of *S. avenae* differed between the six wheat varieties (Table 2). Longzhong No.3 had the highest *S. avenae* WG (777.42 μg), while the least (622.98 μg) weight gain of *S. avenae* occurred on Yong Liang 15. *S. avenae* WG in Yongliang No.15 was significantly lower than in Longzhong No.3 (F_1,4_ = 6.38, *p* < 0.05), whereas WG in the other four cultivars did not differ significantly (F_3,8_ = 5.43, *p* > 0.05) (Table 2).

The mean relative growth rate (MRGR) of all cultivars tested differed from one another. Yongliang No.15 had the lowest MRGR (0.395) and was not significantly different (F_4,10_ = 4.14, *p* > 0.05) from the other four cultivars tested: Longzhong No.3, Huining No.21, Ganchun No.29, and Ganchun No.18. While Longchunza No.2 had the highest mean relative growth rate (0.433), its MRGR was significantly higher (F_3,8_ = 5.45, *p* < 0.05) than Huining No.21, Yongliang No.15, and Ganchun No.18. However, there were no significant differences (F_2,6_ = 1.22, *p* > 0.05) between Longchunza No.2, Longzhong No.3, and Ganchun No.29 (Table 2).

### 3.3. Total Phenol Compound Content

We focused on variation in total phenol and flavonoid content of plants to investigate the effects of *S. avenae* on secondary metabolite accumulation. The total phenol content accumulated varied across all varieties. After 3 days of infestation, total phenol content was higher in Longzhong No.3, Longchunza No.2, and Ganchun No.18. The highest total phenol content was found in Ganchun No.18 (0.947 mg^−1^ g), while Huining No.21 and Yongliang No.15 showed a significantly (F_5,12_ = 617.73, *p* < 0.05) lower accumulation of total phenol content. After 7 d of infestation, four out of the six tested wheat cultivars (that is, Ganchun No.29, Longchunza No.2, Yongliang No.15, and Ganchun No.18) showed an increasing trend. Yongliang No.15 had the highest total phenol content (0.611 mg^−1^ g) when compared to the other cultivars. Longzhong No.3 had a significantly lower total phenol content (−0.250 mg^−1^ g), but it did not differ significantly from Huining No.21 (−0.113 mg^−1^ g). Ten days after infestation, we discovered that Yongliang No.15 had the highest (1.039 mg^−1^ g) and most significant (F_5,12_ = 86.66, *p* < 0.05) increase when compared to others, followed by Huining No.21 (0.287 mg^−1^ g) and Ganchun No.29 (0.148 mg^−1^ g). It was also observed that total phenol content decreased in Longchunza No.2 and Ganchun No.18 (−0.127 and −0.218 mg^−1^ g respectively) compared with 3 and 7 d after infestation. Furthermore, there was a statistical trend toward an increase in the total phenol in Ganchun No.29 after 10 days (0.148 mg^−1^ g), which is similar but is clearly inferior to infestation after 7 days (0.372 mg^−1^ g) of infestation. Total phenol content in Huining No.21, on the other hand, increased 10 days after infestation (Figure 2A).

### 3.4. Flavonoid Content

Considering the difference in content of total phenol in infested plants across all cultivars, changes in flavonoid content were also determined to observe its response to *S. avenae* infestation. Changes in flavonoid content in each variety were assessed 3, 7, and 10 days after infestation. After 3 days of infestation, flavonoid content increased in three cultivars, namely, Huining No.21, Yongliang No.15, and Ganchun No.18. There was a significant difference (F_2,6_ = 2801.95, *p* < 0.05) between the three cultivars within the same time period. Ganchun No.18 had the highest increase in flavonoid content (2.730 mg^−1^ g), followed by Yongliang No.15 (0.909 mg^−1^ g), and Huining No.21 (0.528 mg^−1^ g). Interestingly, at 7 days after infestation, the flavonoid content decreased in five cultivars with the exception of Yongliang No.15, which had the highest flavonoid accumulation (1.274 mg^−1^ g). Huining No.21 had the least flavonoid accumulation (−1.173 mg^−1^ g). It is also worth noting that there were no significant differences in flavonoid content changes between Huining No.21 and Longzhong No.3, or between Huining No.21 and Ganchun No.29. At 10 d after infestation, significant increases in flavonoid content were observed in Yongliang No.15, similar to the changes observed at 3 and 7 d. Yongliang No.15 (1.039 mg^−1^ g) had a significantly (F_2,6_ = 53.18, *p* < 0.05) higher increase in flavonoid content than Huining No.21 (0.287 mg^−1^ g) and Ganchun No.29 (0.148 mg^−1^ g). Huining No.21 and Ganchun No.29, on the other hand, showed an increasing trend when compared to 7 d. Longzhong No.3 had the greatest decrease in flavonoid content (−0.497 mg^−1^ g) when compared to Longchunza No.2 and Ganchun No.18 at 10 d (Figure 2B). Interestingly, we noticed that the trend in flavonoid content change was very similar to that previously observed in the total phenol at 10 days (Figure 2A,B).

### 3.5. Net Changes of Total Phenol Content and Flavonoid Content

The total phenol and flavonoid content accumulation varied significantly across all cultivars at 3, 7, and 10 days after infestation. A similar trend of total phenol and flavonoid content accumulation was observed across all cultivars 10 days after infestation. The overall trend of phenol and flavonoid accumulation varied among the six tested wheat varieties at 3 and 7 days after infestation. As a result, we calculate the net changes in total phenol and flavonoid content in the six wheat varieties tested. Figure 3 depicts the net changes in total phenol content and flavonoid content in the six wheat varieties. Noticeably, except for Longchunza No.2, the trend in change of total phenol content was consistent with the flavonoid content across all cultivars. Meanwhile, Yongliang No.15 had the highest increase in total phenol content (1.155 mg^−1^ g) and flavonoid content (3.819 mg^−1^ g), followed by Ganchun No.18. Conversely, Longzhong No.3 had the lowest phenol (−0.572 mg^−1^ g) and flavonoid content (−3.239 mg^−1^ g) (Figure 3).

### 3.6. Correlation between Wheat Resistance to S. avenae and Biological Parameters of Wheat Aphid and Accumulation of Secondary Metabolites

To better elucidate the association of wheat resistance to *S. avenae* with accumulation of secondary metabolites, Pearson’s correlation coefficient (r) was used to determine the correlation between the change in total phenol and flavonoid content and AQR, MRGR, DD, and WG (Figure 4). Pearson’s correlation coefficient (r) is a measure of the linear relationship between two variables. If the value is greater than 0.8, there is a strong positive correlation. Pearson’s correlation analysis revealed a significant (*p* ≤ 0.01) and strong positive correlation (*r* = 0.87) between total phenol content and flavonoid content accumulation. AQR, an important index for assessing resistance to *S. avenae*, was found to be significantly correlated positively with MRGR (*p* ≤ 0.01, *r* = 0.68) and weight gain (*p* ≤ 0.05, *r* = 0.53). AQR, on the other hand, was negatively correlated with DD (*p* ≤ 0.05, *r* = *−0.50*), total phenol content, and flavonoid content accumulation (Figure 4). The correlation between MRGR and WG was significant (*p* ≤ 0.05, *r* = −0.50), but there was a significant negative correlation with DD (*p* ≤ 0.01, *r* = −0.65) and flavonoid content accumulation (*p* ≤ 0.01, *r* = −0.54). We also discovered a statistically significant (*p* ≤ 0.05) negative correlation between WG and the change in total phenol (*r* = −0.57) and flavonoid content (*r* = −0.58).

## 4. Discussion

For decades, identifying wheat varieties with resistance against aphid attack has been a major focus of research. Several studies show that antibiosis is a more important mechanism in wheat resistance to aphid than antixenosis and tolerance [34,35,36]. Developmental duration (DD), weight gain (WG), mean relative growth rate (MRGR), nymph survival rate, fertility (F), and intrinsic rate of increase (r_m_) are the basic biological parameters for evaluating and understanding insect population dynamics [37]. These are commonly used to evaluate insect adaptability, phenotypic plasticity, population dynamics of insect response changes in environmental conditions, and insect resistance level in host plants [37]. Indeed, many published studies favored using the DD, WG, and MRGR to assess the plant resistance to insect attack [38,39,40,41,42].

The first question in this study was to assess wheat resistance to *S. avenae* by observing AQR and biological parameters of *S. avenae*. In this study, Ganchun No.18 had the longest *S. avenae* DD and lowest AQR, whereas Longchunza No.2 had the shortest DD and the highest AQR. Antibiosis primarily inhibits the target pest’s life cycle, including development, reproduction, and survival [43]. This suggests that the Ganchun No.18 cultivar may have possessed a stronger antibiosis trait that inhibited the growth of *S. avenae*. This was confirmed by negative correlation of AQR with DD, which hampered the development of *S. avenae*. Longchunza No.2’s susceptibility to *S. avenae* is demonstrated by the absence of the antibiosis trait. This finding is consistent with Fan and Lan’s previous observations that the DD of *S. avenae* in susceptible varieties (lines) was shorter than in resistant ones [44,45]. In comparison to the other cultivars, Yongliang No.15 had the lowest MRGR and WG. The positive correlation of AQR with MRGR and WG supported this. Fan et al. (2020) reported a similar result [44].

Plants, inevitably, are attacked by diverse phytophagous insects in agricultural and natural environments. As a result, plants have evolved a plethora of chemical defenses, such as secondary metabolites, to prevent or reduce the damage caused by arthropod herbivores [14]. Phenol and flavonoid compounds, common plant secondary metabolites, may affect insect herbivores directly (weight, developmental rate, and survivorship) and/or indirectly (by modifying insect vulnerability to parasitoids and pathogens) through their toxicity [46,47,48]. Among secondary metabolites, it is well established that the accumulation of phenol substances is the most visible in plants following insect herbivore attack [49]. Nonetheless, the link between total phenol compounds and insect performance remains controversial. Correlations between total phenol compounds and insect performance have ranged from negative to zero, but positive correlations have also been reported [47]. We next addressed the second question: the effects of *S. avenae* infestation on total phenol and flavonoid content in different wheat varieties after 3, 7, and 10 d of infestation. Total phenol and flavonoid content accumulation varied across all cultivars in this study. Change in phenol and flavonoid content is a dynamic process as aphid feeding time increases. For example, the total phenol and flavonoid content of Longzhong No.3, Longchunza No.2, and Ganchun No.18 decreased over time. Ganchun No.18 had the highest total phenol and flavonoid content. Plant phenol and flavonoid contents often change in response to insect herbivore attacks [23,24,49]. Previous research has shown that higher levels of phenols in plants can prevent pests from feeding. In comparison to the other cultivars, this suggests that Ganchun No.18 appears to be able to accumulate sufficient phenol and flavonoid content in response to wheat aphid attack. In accordance with the current findings, Zhu et al. (2011) discovered that the total phenol content of resistant varieties was significantly increased with the length of infested time, but there were significant differences in the extent of increase among different varieties [50]. Green et al. (2003) also observed that the contents of some phenol compounds differed between pigeonpea, *Cajanus cajan* varieties, as did their sensitivity to podworms, *Helicoverpa armigera* [51].

The third question was whether there was a relationship between AQR, biological parameters, and accumulation in total phenol content and flavonoid content in the six wheat varieties. The accumulation of total phenol and flavonoid content correlated positively with developmental duration (DD), whereas MRGR and WG correlated negatively. This suggests that the high accumulation of total phenol and flavonoid contents hampered the performance of the *S. avenae* by delaying its development. This finding is consistent with that of Goławska and Lukasik (2009), who also discovered an inverse relationship between phenol levels and aphid abundance and biology [25].

Despite the fact that there was a significant and strong positive correlation between an increase in total phenol content and an increase in flavonoid content, flavonoid showed a statistically stronger correlation to biological parameters than total phenol. This finding was similar to those of Wang et al. (2019) [49], who found that the gypsy moth larvae (*Lymantria dispar*) adapted differently to different secondary metabolites of poplar [49]. This also explains why, despite having the highest AQR, Longchunza No.2 has the highest total phenol content. It also suggests that Ganchun No.18 and Yongliang No.15 may be resistant to *S. avenae.* This was supported by their high accumulation of phenol and flavonoid contents, and *S. avenae*’s low MRGR. As a result, these cultivars can be used in areas where *S. avenae* poses a significant threat to wheat production. Given that the focus of the study was on flavonoid and phenol response of wheat to *S. avenae* infestation, Piesik et al. found that plants also can release specific VOCs (terpenes, fatty acid derivatives, benzenoids) in response to insect attack [52,53]. Thus, further studies are necessary to explore the function of volatile organic compounds (VOCs) in wheat in response to *S. avenae* attacks.

## 5. Conclusions

In summary, this study evaluated the level of resistance in six wheat varieties to *S. avenae* using AQR and biological parameters. It also assessed the relationship between AQR, biological parameters, and total phenolic and flavonoid content accumulation. The results show that Ganchun No.18 and Yongliang No.15 were resistant to *S. avenae*. In contrast, Longchunza No.2 was more susceptible to *S. avenae* than the other cultivars and performed poorly against *S. avenae*. The total phenol and flavonoid content accumulation correlated positively with developmental duration (DD), whereas MRGR and WG correlated negatively. Furthermore, flavonoid had a stronger relationship with biological parameters than total phenol. In areas where wheat aphids are a major concern, Ganchun No.18 and Yongliang No.15 cultivars can be used.

## Figures and Tables

**Figure 1 insects-13-00508-f001:**
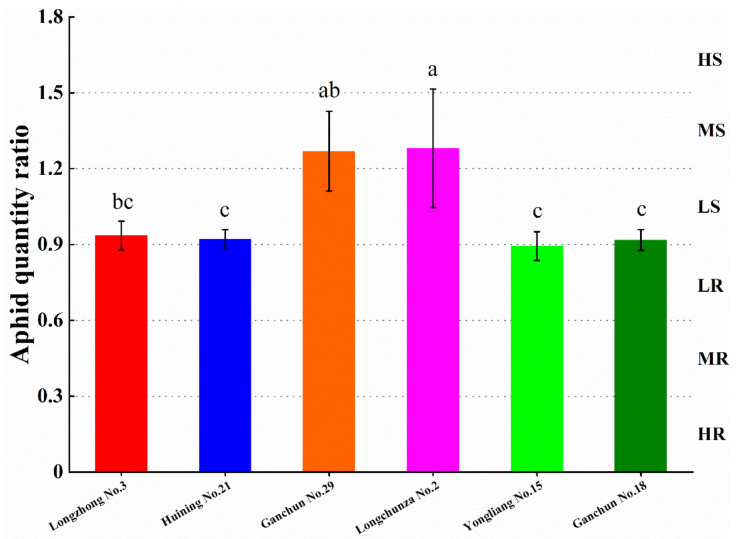
The aphid quantity ratio (AQR) and resistance level of *S. avenae* on six wheat varieties. Values represent the mean ± SD (*n* = 3). HR stands for highly resistant; MR stands for moderately resistant; LR stands for low resistance; LS stands for low susceptible; MS stands for moderately susceptible; HS stands for highly susceptible. Different lowercase letters above the columns indicate significant differences among different cultivars at the 0.05 level (Tukey’ test).

**Figure 2 insects-13-00508-f002:**
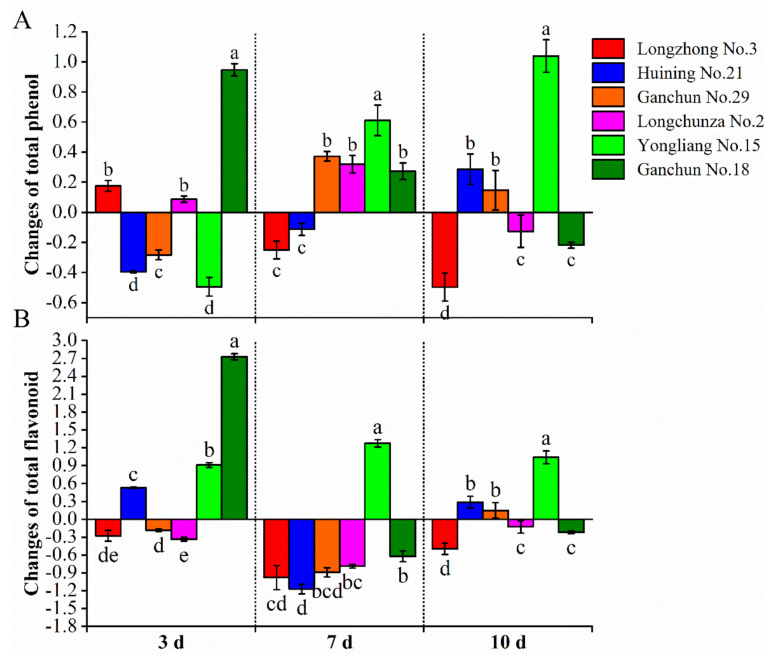
The accumulation effect of *S. avenae* feeding at 3, 7, and 10 days on the (**A**) total phenol content (mg^−1^ g) and (**B**) flavonoid content (mg^−1^ g) in different wheat varieties. The data represented are the mean of three replicates; ±bars indicate standard deviation (SD) of the mean. Different lowercase letters above the columns indicate significant differences among different cultivars at the 0.05 level (Tukey’ test).

**Figure 3 insects-13-00508-f003:**
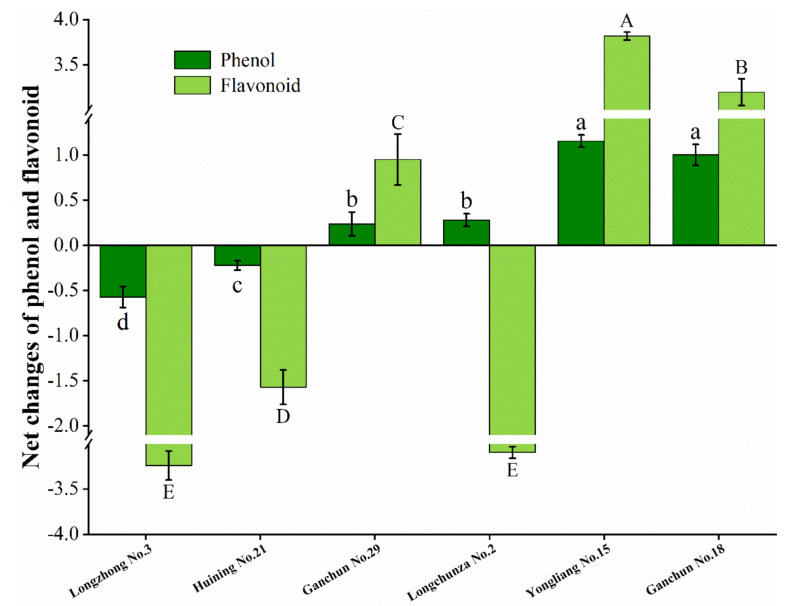
Net changes in total phenol content and flavonoid content in six wheat varieties (mg^−1^ g). The data represented are the mean of three replicates; ±bars indicate standard deviation (SD) of the mean. Different letters above the columns indicate significant differences among different cultivars at the 0.05 level (Tukey’ test).

**Figure 4 insects-13-00508-f004:**
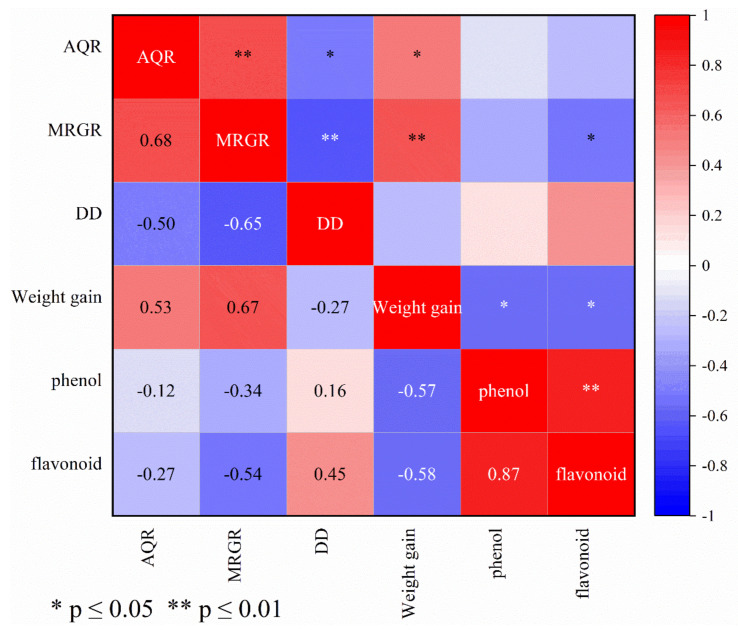
Net changes in total phenol content and flavonoid content in six wheat varieties (mg^−1^ g). AQR shows aphid quantity ratio, MRGR shows mean relative growth rate, and DD shows developmental duration. Color code ranges from blue = strong negative correlation (r = −1) to white = no correlation (r = 0) to red = positive correlation (r = +1).

**Table 1 insects-13-00508-t001:** Classification criterion of wheat resistance to the aphid.

Resistance Level	AQR	Resistance Type
0	0	Immune (I)
1	0.01–0.30	Highly resistant (HR)
2	0.31–0.60	Moderately resistant (MR)
3	0.61–0.90	Low resistant (LR)
4	0.91–1.20	Low susceptible (LS)
5	1.21–1.50	Moderately susceptible (MS)
6	>1.50	Highly susceptible (HS)

**Note:** AQR—aphid quantity ratio.

**Table 2 insects-13-00508-t002:** Biological parameters of *S. avenae* in the six wheat varieties.

Wheat Varieties	Development Duration/d	Weight Gain/μg	MRGR
Longzhong No.3	6.61 ± 0.16 ab	777.42 ± 26.18 a	0.418 ± 0.007 ab
Huining No.21	6.68 ± 0.01 ab	667.51 ± 17.76 ab	0.400 ± 0.012 b
Ganchun No.29	6.54 ± 0.08 ab	760.89 ± 41.41 ab	0.423 ± 0.010 ab
Longchunza No.2	6.19 ± 0.04 b	749.68 ± 63.07 ab	0.433 ± 0.017 a
Yongliang No.15	6.63 ± 0.13 ab	622.98 ± 102.59 b	0.395 ± 0.001 b
Ganchun No.18	6.83 ± 0.40 a	660.50 ± 13.76 ab	0.403 ± 0.015 b

**Note:** Values expressed as mean ± SD; means in the same column followed by different lowercase letters are significantly different at 0.05 level; comparison of development period, weight gain, and *MRGR* using Tukey HSD test; MRGR—mean relative growth rate.

## Data Availability

The datasets in this study are available from the corresponding author on reasonable request.

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
