# Peer review of "Responses of Six Wheat Cultivars (Triticum aestivum) to Wheat Aphid (Sitobion avenae) Infestation"

_insects, 2022, doi:10.3390/insects13060508_

Round 1

Reviewer 1 Report

Below are minor comments

Line 16 and 17: Please be consistent, change number 6 to six in text

Line 59 and 65: change plant resistance to host plant resistance

Line 73: leave a space between attack and citations

Line 77: put a comma between studies and flavonoid. Alternatively, find a joining word as it does not read well.

Line 90-92: For clarity the sentence was rephrased to “This study was conducted to screen the resistance of six wheat cultivars against S. avenae and explore the resistance mechanisms”. Please confirm if the meaning is retained.

Line 105: remove “with” before “were”

Line 111 (2.2 sub-heading): add “the” before laboratory

Line 121 (AQR formula): change cultivars (lines) to cultivar (line)

Line 123: The sentence was rephrased “…as described by Painter [29] and previously used in other studies [30,31]”.

Line 124: change “level are” to “level is”

Line 141: is developmental duration abbreviated as D or DD? Please be consistent. Also, see line 144 and 145.

Line 181: change analyses to analysis

Line 200-205 can be broken down into two. After the mention of two cultivars that were moderately susceptible (MS), cut the sentence and then start a new one. Also write number 6 as six or six (6).

Line 312-321 (correlations): insert a suitable symbol (≤) not <=

Line 347:  developmental duration can be entirely replaced by DD since previously explained

Author Response

Thanks for your valuable suggestions, we have carefully replied to all of your comments.

Reviewer 2 Report

Article ID: insects-1720586 entitled “Responses of six wheat cultivars (Triticum aestivum L.) to wheat aphid (Sitobion avenae) infestation” by Zhang et al. is a well written manuscript that flows well from start to finish with important new data on evaluation of resistant varieties of six wheat cultivars in China. I recommend consideration for publication in INSECTS in present form. I added some minor annotations to the attached PDF for authors’ revision.   

Author Response

(The authors gave the same response as above.)

Reviewer 3 Report

Review ID 1720586

Responses of six wheat cultivars (Triticum aestivum L.) to  wheat aphid (Sitobion avenae) infestation

               This is quite well organized manuscript. I found this “ms” interesting and innovative. However, a few questions must be explained more precisely.

Critical review:

  1. You studied responses of wheat to aphids. In lines 71-80 there is no information about pest insects of stored material. Not acceptable.
  2. Lines 90-92. The aim of the study is poorly presented.
  3. What about volatile organic compounds? Flavonoids and phenols are usually recognized as the second parameter of insect choice. Explain please.
  4. Try to redirect the discussion a bit to volatile compounds. Without it, I don't think the article is complete.

Other papers to add:

Sitophilus granarius responses to blends of five groups of cereal kernels and one group of plant volatiles

Journal of Stored Products Research 62: 36-39 (2015)

DOI: 10.1016/J.JSPR.2015.03.007

Tribolium confusum responses to blends of cereal kernels and plant volatiles

Journal of Applied Entomology 140, 558–563 (2016)

DOI: 10.1111/JEN.12284

Author Response

(The authors gave the same response as above.)

Round 2

Reviewer 3 Report

Accepted.